# FIGHT FIRE WITH FIRE: COUNTERING BAD SHORTCUTS IN IMITATION LEARNING WITH GOOD SHORTCUTS

## ABSTRACT

When operating in partially observed settings, it is important for a control policy to fuse information from a history of observations. However, a naive implementation of this approach has been observed repeatedly to fail for imitation-learned policies, often in surprising ways, and to the point of sometimes performing worse than when using instantaneous observations alone. We observe that behavioral cloning policies acting on single observations and observation histories each have their strengths and drawbacks, and combining them optimally could achieve the best of both worlds. Motivated by this, we propose a simple model combination approach inspired by human decision making: we first compute a coarse action based on the instantaneous observation, and then refine it into a final action using historical information. Our experiments show that this outperforms all baselines on CARLA autonomous driving from images and various MuJoCo continuous control tasks.

| Input Image | BC-SO | BC-OH | Our model |
|---|---|---|---|

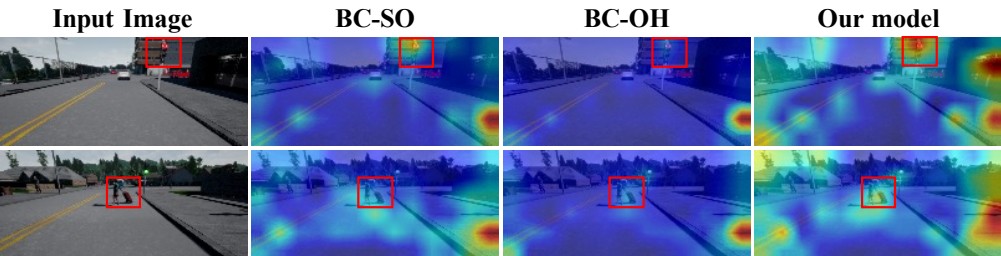

Figure 1: Attention maps of baseline imitation methods and our method, depicted on the CARLA driving task. Behavioral cloning from single observation (BC-SO) attends to the appropriate visual cues in the scene (the traffic light and pedestrian), but performs suboptimally due to lack of information. Behavioral cloning from observation history (BC-OH) has access to all required information, but manifests the "copycat problem"; it relies excessively on extrapolating past actions, and fails to attend to important visual cues. Our "coarse-to-fine" imitator combines the advantages of each method, and outperforms both.

## 1 INTRODUCTION

Learning complex decision behaviors such as driving is a challenging task for machine learning models. If given expert demonstrations, imitation learning reduces the complex policy learning to supervised learning by mimicking the expert's behavior, which is more sample efficient and has more human-like behavior than the reinforcement learning methods. Many existing imitation learning methods employ the behavioral cloning (BC) strategy to directly regress the mapping from the observations to the expert actions. Comparing with other families of imitation learning such as GAIL (Ho & Ermon, 2016; Zhang et al., 2020) and DAGGER (Ross et al., 2011; Spencer et al., 2021), behavioral cloning does not have access to online interaction or queryable experts, resulting in severe distributional shift due to the compounding errors during online testing.

We are interested in a specific type of distributional shift in the POMDP setting – copycat problem (Wen et al., 2020). Because of the partial observation, it is common to take the observation

history along with the current observation as the input of the imitation policy to compensate for the missing information. However, many previous works find that behavioral cloning from observation history performs worse than expected (Muller et al., 2006; Wang et al., 2019; Bansal et al., 2019; de Haan et al., 2019). Wen et al. (2020; 2021) notice that the imitator tends to cheat by simply copying the previous actions rather than taking actions according to the observation and they call it *copycat problem*. This phenomenon has been reported in several papers in different forms, such as past motion cheating (Bansal et al., 2019), inertia problem (Codevilla et al., 2019), causal confusion (de Haan et al., 2019) and etc. All of them are essentially the same: the deep neural networks tend to take the incorrect shortcut from the previous action implied in the observation history, which is much simpler than mining real decision logic from massive amounts of data.

This problem is further illustrated in Figure 1. In this scenario, behavioral cloning from single observation (BC-SO) correctly attends to the traffic light and pedestrian. However, without access to the current speed, it cannot take the accurate brake value to slow down in time. In contrast, behavioral cloning from observation history (BC-OH) pays much less attention to the pedestrian and even ignores the traffic light. Instead, it extrapolates the previous actions, which results in the running red light and collision. In summary, BC-SO has a hard time predicting accurate enough actions but is copycat-free, while with access to more information, BC-OH is able to output more fine-grained actions but suffers from the copycat problem.

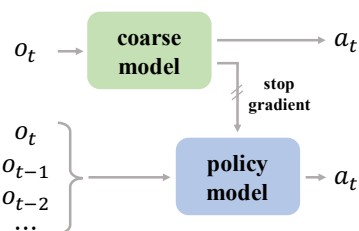

Figure 2: Our approach.

Since both BC-SO and BC-OH have their strengths and drawbacks, it is natural to think about how to combine them in an optimal way that achieves the best of both worlds. Motivated by human decision-making, we propose a coarse-to-fine imitation learning approach, as shown in Figure 2.

We demonstrate our method in four environments, ranging from autonomous driving to robotics control. Our method outperforms other methods (Bansal et al., 2019; Wen et al., 2020; 2021; Ross et al., 2011). Furthermore, we conduct extensive ablation studies to verify our hypothesis for why our approach works.

## 2 RELATED WORK

**Imitation Learning.** Imitation learning is a powerful technique to learn complex decision and control behaviors from observed expert demonstrations. We mainly focus on the widely used behavioral cloning paradigm which directly learns a mapping from observations to expert actions. Behavioral cloning usually suffers from distributional shift because the small error between imitator and expert trajectories will compound along the time and finally leads to encountering unfamiliar states not covered by demonstrations. Several existing works have been proposed to resolve this issue, but most of them either require environmental interactions (Ho & Ermon, 2016; Brantley et al., 2020; Dadashi et al., 2020) or queriable online experts (Ross et al., 2011; Laskey et al., 2017b; Sun et al., 2018; Spencer et al., 2021). Solving the distributional shift problem under the offline setting is still appealing. In this paper, we focus on resolving a specific aspect of distributional shift – copycat problem (Wen et al., 2020), caused by the nature of MDP that the actions are highly correlated along the time. We will discuss the works related to copycat problem in Section 3.2.

**Shortcut Learning.** With the rapid development and widespread use of deep neural networks in academia and industry, more and more limitations of DNN have come into focus. In computer vision and NLP, etc., researchers have found that DNN usually tends to attend to some easy but incorrect clues to make the prediction, such as object texture (Geirhos et al., 2019a), image background (Beery et al., 2018) and word length (Niven & Kao, 2019; McCoy et al., 2019). Geirhos et al. (2020) summarize this phenomenon as shortcut learning, i.e. the DNNs prefer to learn the easier solution (shortcut) rather than taking more effort to learn the intended solution. We regard the copycat problem as a specific instance of shortcut learning problem in imitation learning, and propose a simpler and more desirable shortcut to alleviate the copycat shortcut.

## 3 PRELIMINARIES

We study behavioral cloning algorithms in a partially observed Markov decision process (POMDP). In POMDP, the environment provides an observation $o_t$, which only partially represents the actual state because of noise or sensor occlusion, and a reward $r_t$. To deal with the missing or noisy information, it is common to take the previous few observations $(o_{t-H}, H = 1, 2, \cdots)$ into account (Murphy, 2000; Bansal et al., 2019; Wen et al., 2020; 2021), which forms the observation history $\tilde{o}_t = [o_{t-H}, \cdots, o_t]$. The final goal of policy learning is to train a model $\pi_\theta$ which takes in $\tilde{o}_t$ and outputs the action $a_t$ to maximize the return, i.e. the sum of the rewards $R = \sum_t r_t$.

### 3.1 BEHAVIORAL CLONING

Behavioral cloning (BC) assumes that we are given offline demonstrations $\mathcal{D} = \{(\tilde{o}_t, a_t)\}_{t=1}^N$ collected by an expert $\pi_e$, and no reward values are provided. The best way to maximize the test return is to just mimic the expert and behave as similar to the expert as possible during online testing. So the behavioral cloning reduces policy learning to a simple supervised learning paradigm with a training dataset $\mathcal{D}$. Specifically, BC methods usually minimize the following behavioral cloning loss, where $\mathcal{L}(\cdot, \cdot)$ is usually mean square error (MSE) or negative log likelihood (NLL) loss:

$$\theta^* = \arg\min_\theta \mathcal{L}(\pi_\theta(\tilde{o}_t), a_t) \tag{1}$$

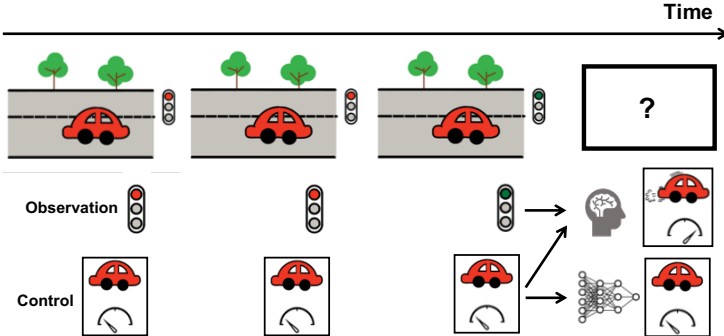

Figure 3: An example of the copycat problem in an autonomous driving scenario. The copycat policy learns to cheat by extrapolating from the previous action. Therefore, when the traffic light turns green, the copycat policy ignores the traffic light and copies the previous action, i.e. staying stationary. On the contrary, the human driver gets the "startup" concept by observing the traffic light and takes the accurate throttle according to the speed.

### 3.2 COPYCAT PROBLEM

In partially observable settings, it is a common practice in control theory (Ang et al., 2005; Welch et al., 1995) and reinforcement learning (Mnih et al., 2015; Schulman et al., 2017) to use historical observations to learn more accurate actions. Similarly, people would expect BC with observation history $\tilde{o}_t$ (BC-OH) to perform better than the model trained from single observation $o_t$ alone (BC-SO). In practice. however, BC-OH could fail unexpectedly. In particular, BC-OH could have lower offline validation loss than BC-SO but performs worse during open-loop testing. As described in Figure 3, Wen et al. (2020) coined the term "copycat problem" to describe this phenomenon: due to the nature of the MDP, i.e. the expert actions are temporally correlated, it is common that the imitator tends to take a shortcut and outputs decisions based on previous actions inferred from historical observations. Many recent works (Wang et al., 2019; de Haan et al., 2019; Wen et al., 2020; Muller et al., 2006; Bansal et al., 2019; Codevilla et al., 2019) report the same findings and try to address this discrepancy in various ways. de Haan et al. (2019) purposes to learn a policy for each possible causal graphs, then perform target interventions to search for the correct one. Bansal et al. (2019) introduces a random dropout on the observation history, Codevilla et al. (2019) introduces a speed prediction regularization module and Wen et al. (2021) treats it as a data imbalance problem and up-weights the training loss of changepoints.

## 4 METHODOLOGY

Previous works address this copycat problem either from a causality (de Haan et al., 2019) or data distribution (Bansal et al., 2019; Codevilla et al., 2019; Wen et al., 2021) point-of-view. Instead we take a much simpler view by analyzing the strengths and drawbacks of BC-SO and BC-OH. We propose to utilize a model combination approach to get the best of both models. Motivated by human decision making, we propose a simple but effective combination approach by predicting a coarse action based on the instantaneous observation (BC-SO), and then refine it into a final action using historical information (BC-OH).

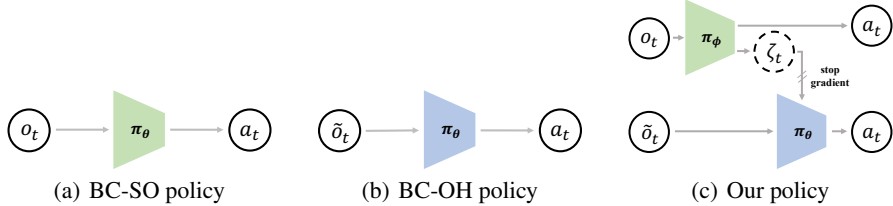

|              (a) BC-SO policy                (b) BC-OH policy                (c) Our policy              |

Figure 4: The comparison of BC-SO, BC-OH and our model. BC-SO and BC-OH learn the mapping from the current observation $o_t$ and the observation history $\tilde{o}_t$ respectively to the expert action $a_t$. Our model combines them together to train a coarse action model $\pi_\phi$ from $o_t$ to $a_t$ and a policy model $\pi_\theta$ from $\tilde{o}_t$ to $a_t$. We extract the coarse action $\zeta_t$ from $\pi_\phi$ (we simply set the action output of $\pi_\phi$ as $\zeta_t$) and then feed it into $\pi_\theta$ by concatenating it with the intermediate feature of $\pi_\theta$.

### 4.1 COMPARISON BETWEEN BC-SO AND BC-OH

As shown in Figure 4(a) and Figure 4(b), BC-SO and BC-OH take input observations of different time horizons. In a partially observed setting, BC-SO does not have access to historical observations to infer its current action. Thus, it is often hard for BC-SO to output smooth actions. However, this also makes BC-SO free from the copycat problem. As shown in Figure 1, BC-SO learns to focus on the traffic light and pedestrian. Therefore, BC-SO excels at making coarse decisions during testing, like starting the vehicle in front of traffic lights or avoiding the pedestrians. Its limitation is that with only the current observation as input, it is hard for BC-SO to reason about its current speed. So it has a hard time driving smoothly on the road, which leads to collisions due to high speed at times.

In contrast, BC-OH takes the observation history as input, so the output action is smoother than BC-SO for most of the time. However, as shown in Figure 1, BC-OH fails to focus on the traffic light or pedestrians during test time. The inclusion of historical information leads to the copycat problem that in these critical scenarios, BC-OH fails to focus on important visual cues but predicts actions solely by extrapolating the previous actions. We observe that, comparing with BC-SO, BC-OH can easily infer its speed from observation history and adjust throttle when the speed is too high. But as illustrated in Figure 3, BC-OH often fails to start the car because it infers the previous actions from the historical observations and then simply copies it.

Both models have strengths and drawbacks and combining them optimally could take the best of them and outperform either individual model alone. According to the model combination approaches in machine learning (Wolpert, 1992; Breiman, 1996; Freund & Schapire, 1997), our solution is motivated by how to combine BC-SO and BC-OH to predict the precise actions according to the observation history while avoiding the copycat problem.

### 4.2 COARSE-TO-FINE IMITATION LEARNING

How might we combine BC-SO and BC-OH to enable relying on historical observations but still avoid the copycat problem? Motivated by human decision making, we propose a coarse-to-fine imitation learning approach, as shown in Figure 4(c). Previous researches have found that there exists a *coarse* action (usually called option or decision in psychology literature) in human's control process (Klein et al., 1995; Johnson & Raab, 2003; Keller & Ho, 1988; Ward et al., 2011). This coarse action is a representation that plays the implicit or explicit psychological role of a candidate for action.

Furthermore, it is generated instantaneously from direct perception, without elaborate cognitive processing (Raab & Johnson, 2007; Raab et al., 2009). Then, humans refine this coarse action into final actions based on more deliberative cognitive processes involving memory and knowledge.

Inspired by this, we design a coarse-to-fine imitator, shown in Figure 4(c). At test time, our model operates in two stages: analogous to BC-SO, our "coarse model" $\pi_\phi$ computes a coarse action $\zeta_t$ based on the instantaneous observation alone, i.e. $\zeta_t = \pi_\phi(o_t)$. Then, a second "policy model" $\pi_\theta$ refines $\zeta_t$ into a final action $a_t$, using additional information from observation histories, i.e. $a_t = \pi_\theta(\tilde{o}_t, \zeta_t)$.

How are these trained? We jointly train the coarse model $\pi_\phi$ and policy model $\pi_\theta$ end-to-end by BC losses in Eq. 2.

$$
\begin{aligned}
\phi^* &= \arg\min_\phi \mathcal{L}(\pi_\phi(o_t), a_t) \\
\theta^* &= \arg\min_\theta \mathcal{L}(\pi_\theta(\tilde{o}_t, \zeta_t), a_t)
\end{aligned}
\tag{2}
$$

In detail, we concatenate the $\zeta_t$ with the intermediate feature of $\pi_\theta$ and feed them into the following layers together. To avoid the historical information leaking into $\pi_\phi$, we isolate the two models with a stop-gradient. The imitator will take advantage of the coarse action and focus on refining it, which avoids taking completely wrong actions by extrapolating from the previous action. The coarse-to-fine process is essentially mimicking the human control process. Recall the vehicle starting example in Section 3.2. The coarse model $\pi_\phi$ takes in the current observation where the traffic light turns green, so it outputs a coarse action $\zeta_t$ representing that the vehicle should decide to start up, i.e. the throttle value should be larger than 0. And then $\pi_\theta$ gets the coarse action and takes an accurate throttle value according to the current speed. With such architecture, our imitation policy will not be affected by the copycat problem and the performance improves a lot, which will be shown in Section 5.

### 4.3 WHY SHOULD COARSE-TO-FINE IMITATION WORK?

Note that in the scheme described above, the coarse action $\zeta_t$ is inferred from the instantaneous observation $o_t$, which in turn is contained inside the observation history $\tilde{o}_t$. Thus, by the data processing inequality, $\zeta_t$ can not add any new information to the policy model $\pi_\theta(\cdot|\tilde{o}_t, \zeta_t)$. Without the $\zeta_t$ input, however, the policy model $\pi_\theta$ reduces to a plain BC-OH policy. This begs the question: why should coarse-to-fine imitation produce any better results than BC-OH? Yet, as we will show in our experiments, it does so consistently across many environments, and often by dramatic margins! We argue that this may be understood from the perspective of optimization difficulties, which lie at the heart of the copycat problem (de Haan et al., 2019; Wen et al., 2020; 2021).

It has previously been shown that widely used neural network training algorithms prefer simple solutions (Valle-Perez et al., 2019; Shah et al., 2020; Huh et al., 2021). While such simple solutions are often preferable, simple solutions exploiting spurious correlations in training data can fail catastrophically under distributional shift (Geirhos et al., 2020; 2019b; Beery et al., 2018). Copycat BC-OH solutions are an example of such a shortcut: they tend to rely solely on information about previous actions contained in the observation history, since this "simple" solution suffices to achieve low behavioral cloning losses, due to temporal correlations in expert training data.

Previous solutions aim to counter this shortcut-learning tendency in the imitation network, such as by upweighting changepoints to preferentially penalize copycats (Wen et al., 2021), or by adversarially removing past action information in the observation history (Wen et al., 2020). We take a different track. To compete with the copycat shortcut above, we provide a second viable "BC-SO shortcut": simply copy the action produced by BC-SO. Recall that BC-SO produces meaningful, but coarse solutions that are responsive to environmental observations (unaffected by past actions), but crippled by a lack of historical information. As such, directly copying the BC-SO solution suffices to produce quite low BC losses. Further, the BC-SO shortcut is arguably even simpler than the copycat shortcut which involves first recovering past action information from the observation history and then copying it.

Finally, we argue that the BC-SO shortcut is also relatively desirable compared to the copycat shortcut. First, since BC-SO does necessarily attend to important elements of the current observation, our final policy $\pi_\theta$ cannot easily ignore this information. Second, with training, $\pi_\theta$ may learn to correctly use the additional historical information in $\tilde{o}_t$ to improve further over the BC-SO shortcut when needed.

# 5 EXPERIMENTS

In our experiments, we aim to first verify that our model can resolve the copycat problem and improve over BC-OH as well as previously proposed solutions. Next we will verify the following key hypotheses about why and how our approach works: **H1 (simplicity of BC-SO shortcut):** To effectively counter the copycat shortcut, it is important to set up the BC-SO shortcut to be as "simple" as possible; **H2 (combination correctness):** It is not sufficient to combine BC-SO and BC-OH in other ways. Providing the BC-SO shortcut to BC-OH is important; **H3 (causality correctness):** After training, the final coarse-to-fine imitator learns the correct causal relation and is able to effectively switch between the BC-SO shortcut solution and utilizing historical information.

## 5.1 EXPERIMENT SETUP

We evaluate our method on driving simulator CARLA (Dosovitskiy et al., 2017), and three OpenAI Gym MuJoCo (Todorov et al., 2012; Brockman et al., 2016) robotics control environments.

**CARLA.** CARLA is a photorealistic urban autonomous driving simulator (Dosovitskiy et al., 2017), and is a commonly adopted testbed for imitation learning (Codevilla et al., 2018; 2019; Chen et al., 2020; Wen et al., 2021; Prakash et al., 2021). We use the CARLA100 dataset (Codevilla et al., 2019) to train all methods, containing 100 hours of driving demonstrations. We evaluate all methods in the hardest CARLA benchmark, *Nocrash-Dense*, which has the most number of pedestrians and heaviest traffic. As discussed in Wen et al. (2021), to construct a pure POMDP, we only use the sequential frames as input without the current speed. We train all the methods three times from different random initializations to account for the high variance (Codevilla et al., 2019). We report the mean and standard deviation of four metrics: %success, %progress, #collision and #timeout. The %success is the number of episodes that are fully completed out of 100 pre-designed evaluation routes. The %progress refers to the fraction of the distance traveled by the agent towards a goal location. The #collision counts the times that the agent crashes into other vehicles, pedestrians, and obstacles. And the #timeout counts the times that the agent fails to reach the destination despite no collision within the specified time, usually caused by unsuccessful starts, wrong routes or traffic jams.

We train all approaches with the CILRS (Codevilla et al., 2019) backbone. We set the length of observation history to $H = 7$ and stack the sequential frames along the channel dimension as the model input like what Bansal et al. (2019); Wen et al. (2021) do. For our method, the whole model consists of two modules, a coarse network $\pi_\phi$ with $o_t$ as input and one policy network $\pi_\theta$ with $\tilde{o}_t$ as input. And we use late-fusion by concatenating the output of $\pi_\phi$, i.e. the decision $\zeta_t$, with the feature of the penultimate fully-connection layer of $\pi_\theta$ through a stop-gradient layer. We train both modules end-to-end. See Supp for the architecture and training details.

**Mujoco.** We evaluate our method in three standard OpenAI Gym MuJoCo continuous control environments: Hopper, Ant and HalfCheetah. We generate expert data from a PPO (Schulman et al., 2017) policy (10k samples for Ant and Walker2D, and 20k for Hopper). To simulate partially observations, We add Gaussian noise $N(0, \sigma^2)$ with $\sigma = 0.2$ to joint velocities. We stack $H = 2$ frames to form the observation history. We train all methods three times with different random initializations and report the final evaluation rewards with their mean and standard deviation. See Supp for network architecture and training details.

We extensively compare our method with baseline methods: **BC-SO**, **BC-OH**, **Fighting-Copycat-Agents (FCA)** (Wen et al., 2020), **KeyFrame** (Wen et al., 2021), **History-Dropout** (Bansal et al., 2019), **Average-Ensemble** and **DAGGER** (Ross et al., 2011). See Supp for the description and implementation of each method.

## 5.2 IMITATION RESULTS

**CARLA.** The evaluation results on *Nocrash-Dense* benchmark are shown in Table 1. We can see that according to %success, %progress and #collision, BC-OH performs significantly better than BC-SO because BC-SO does not have access to the additional information provided by the observation history. The most common failure case of BC-SO is that the car is accelerating when going straight because it cannot infer the current speed from a single observation, contributing to an extremely high #collision. This does not happen on BC-OH because it can easily adjust its speed according to the

Table 1: CARLA *Nocrash-Dense* results.

| METHOD | %SUCCESS ($\uparrow$) | %PROGRESS ($\uparrow$) | #COLLISION ($\downarrow$) | #TIMEOUT ($\downarrow$) |
|---|---|---|---|---|
| BC-SO | 13.1 ± 1.8 | 40.8 ± 3.0 | 76.4 ± 3.5 | **11.1 ± 2.9** |
| BC-OH | 34.1 ± 7.5 | 62.2 ± 9.4 | **30.2 ± 7.9** | 36.1 ± 14.5 |
| OURS | **49.3 ± 3.6** | **75.0 ± 1.6** | 39.4 ± 5.0 | 12.0 ± 1.9 |
| FCA | 31.2 ± 5.2 | 66.5 ± 4.1 | 34.4 ± 8.1 | 35.3 ± 9.6 |
| KEYFRAME | 41.9 ± 6.2 | 70.2 ± 4.0 | 33.9 ± 6.6 | 24.8 ± 7.9 |
| HISTORY-DROPOUT | 35.6 ± 3.5 | 67.0 ± 2.7 | 45.3 ± 3.5 | 20.3 ± 5.6 |
| AVERAGE-ENSEMBLE | 41.7 ± 3.1 | 71.5 ± 3.2 | 43.7 ± 4.0 | 15.0 ± 0.8 |
| DAGGER 150K | 42.7 ± 5.7 | 71.3 ± 1.9 | 35.0 ± 3.6 | 23.0 ± 7.1 |

Table 2: MuJoCo rewards.

| METHOD | HOPPER | ANT | HALFCHEETAH |
|---|---|---|---|
| BC-SO | 589 ± 94 | 4198 ± 433 | 489 ± 77 |
| BC-OH | 628 ± 99 | 2922 ± 1266 | 639 ± 121 |
| OURS | **1124 ± 135** | **4798 ± 304** | **1448 ± 74** |
| FCA | 831 ± 108 | 3727 ± 926 | 1148 ± 81 |
| KEYFRAME | 696 ± 28 | 2930 ± 1321 | 1062 ± 127 |
| HISTORY-DROPOUT | 539 ± 33 | 4069 ± 517 | 1215 ± 70 |
| AVERAGE-ENSEMBLE | 504 ± 47 | 4659 ± 396 | 729 ± 50 |
| DAGGER 8K | 2383 ± 294 | 4097± 418 | 1842 ± 10 |
| PPO EXPERT | 3445 | 5566 | 1941 |

dynamic information. However, the #timeout of BC-OH is the highest among all methods, which is mainly caused by unsuccessful start: When the car is stationary, either stops in front of traffic light or pedestrians, BC-OH often remains stationary until timeout, which is a typical copycat phenomenon. In contrast, the #timeout of BC-SO is pretty low, most of which are caused by traffic jams and wrong routes, indicating that BC-SO is free from the copycat problem.

Our method outperforms all the observation-history-based baselines on %success and %progress, even including DAGGER which benefits from the 150K extra online expert data. Especially, our method significantly reduces the #timeout to 12.0, very close to BC-SO. And very few of these timeout cases are caused by copycat-related issues.

**MuJoCo.** The evaluation results on Hopper, Ant, and HalfCheetah are shown in Table 2. We can see that although BC-OH has historical observations as input, it only performs slightly better than BC-SO on Hopper and HalfCheetah, while performing worse than BC-SO on Ant, due to the copycat problem. Our method easily beats all the naive behavioral cloning baselines. We also compare our method with previous methods that intend to address the copycat problem. Our method outperforms all three of them (FCA, KeyFrame and History-Dropout) in all environments. On both Hopper and HalfCheetah, none of the above-mentioned offline methods beat DAGGER, which intends to be an oracle method with access to online samples.

### 5.3 ABLATION STUDY

To evaluate the effect of each module, we conduct ablation study experiments on CARLA about **1) different fusion stages**: late-fusion (ours), middle-fusion or early-fusion (see Supp for the architecture details); **2) coarse action selections**: use the final action of $\pi_\phi$ or the intermediate features, e.g. Restnet Layer2's feature or Resnet output, as $\zeta_t$; **3) ways to combine BC-SO and BC-OH**: use BC-SO as $\pi_\phi$ and BC-OH as $\pi_\theta$ (ours), BC-OH as $\pi_\phi$ and BC-OH as $\pi_\theta$, or BC-OH as $\pi_\phi$ and BC-SO as $\pi_\theta$; **4) stop-gradient**: with or without stop-gradient; **5) end-to-end vs. two-stage training**: train the $\pi_\phi$ and $\pi_\theta$ end-to-end or two-stage: first train $\pi_\phi$ and use its checkpoint when training $\pi_\theta$. The results are shown in Table 3 and we provide detailed analysis for each ablation respectively in the following paragraphs.

Table 3: CARLA ablation study results.

| METHOD | %SUCCESS ($\uparrow$) | %PROGRESS ($\uparrow$) | #COLLISION ($\downarrow$) | #TIMEOUT ($\downarrow$) |
|---|---|---|---|---|
| OURS | $49.3 \pm 3.6$ | $75.0 \pm 1.6$ | $39.4 \pm 5.0$ | $12.0 \pm 1.9$ |
| EARLY-FUSION | $51.4 \pm 4.0$ | $75.9 \pm 2.1$ | $37.4 \pm 5.1$ | $12.7 \pm 2.6$ |
| MIDDLE-FUSION | $49.6 \pm 1.7$ | $74.8 \pm 1.9$ | $37.3 \pm 4.0$ | $13.3 \pm 3.0$ |
| RESNET LAYER2 AS $\zeta_t$ | $39.8 \pm 4.0$ | $69.4 \pm 1.5$ | $32.6 \pm 1.5$ | $28.1 \pm 4.4$ |
| RESNET OUTPUT AS $\zeta_t$ | $44.8 \pm 5.2$ | $71.9 \pm 4.7$ | $40.0 \pm 5.7$ | $15.9 \pm 3.4$ |
| BC-OH AS $\pi_\phi$, BC-OH AS $\pi_\theta$ | $37.1 \pm 3.4$ | $66.9 \pm 2.8$ | $34.1 \pm 3.4$ | $28.9 \pm 6.0$ |
| BC-OH AS $\pi_\phi$, BC-SO AS $\pi_\theta$ | $31.7 \pm 5.7$ | $62.2 \pm 6.1$ | $28.3 \pm 9.0$ | $40.0 \pm 14.7$ |
| W/O STOP-GRADIENT | $44.4 \pm 5.5$ | $71.4 \pm 3.0$ | $39.2 \pm 4.7$ | $17.1 \pm 2.1$ |
| W/O END-TO-END | $47.3 \pm 3.8$ | $71.2 \pm 1.2$ | $43.4 \pm 5.5$ | $9.8 \pm 3.1$ |

We can see that given the coarse action $\zeta_t$, it makes no difference where to inject it into $\pi_\theta$ (%success ranges from $49.3$ to $51.4$ and other metrics are similar too), indicating that $\zeta_t$ provides a simpler shortcut than inferring and copying the previous action from the observation history. Wherever we inject $\zeta_t$, the neural network prefers to adopt it directly as the decision rather than expending greater effort to take the copycat shortcut.

If we use the intermediate features (Resnet Layer2 and Resnet output) of the decision model as opposed to the output action of $\pi_\phi$, then %success drops from $49.3$ to $39.8$ and $44.8$ respectively. Moreover, comparing the %success and #timeout, we find that the shallower the feature, the lower %success and the higher #timeout, indicating that using shallower features as coarse action is more likely to suffer from the copycat problem. Intuitively, it is harder for $\pi_\theta$ to extract the coarse action from shallower features than from later ones and thus $\pi_\theta$ has less incentive to take the coarse action shortcut compared with the original copycat shortcut. Combining with previous observation in different fusion stages ablation, this experiment clarifies that "simplicity" in **H1 (simplicity of BC-SO shortcut)** focuses more on what is the input rather than where to input to the policy $\pi_\theta$ and indeed, the simpler the better.

If we use different combinations of BC-SO and BC-OH, such as BC-OH as $\pi_\phi$, BC-SO as $\pi_\theta$ or BC-OH as both $\pi_\phi$ and $\pi_\theta$, the imitator performs similarly to BC-OH (%success ranges from $31$ to $37$), indicating that introducing historical information into the coarse model will lead to copycat problem, which verifies **H2 (combination correctness)**, and also verifies that our method's performance is not due to higher model capacity.

Moreover, we find that the agent performs worse if we remove stop-gradient (%success drops from $49.3$ to $44.4$), which is mainly due to timeout (increasing from $12.0$ to $17.1$). The increasing #timeout shows that our method without stop-gradient still suffers from the copycat problem due to historical information leakage during back-propagation. Such a variant without stop-gradient also demonstrates **H2 (combination correctness)**.

And training $\pi_\phi$ and $\pi_\theta$ stage-by-stage, the performance also deteriorates (%success drops from $49.3$ to $47.3$). We can see that its #timeout is even fewer than ours but it gets a significantly higher #collision (#collision increases from $39.4$ to $43.4$), indicating that the agent trained stage-by-stage behaves more like BC-SO, i.e. suffering from less copycat but failing to brake in time. We hypothesize that if we use a pretrained coarse model $\pi_\phi$ at the beginning of $\pi_\theta$ training, $\pi_\theta$ will prefer to just copy the $\zeta_t$ as its output rather than learn from scratch to take accurate actions according to both the coarse action and observation history, which can also be viewed as overly relying on the "simpler" solution, i.e. shortcut learning and sheds light on the importance of **H2 (combination correctness)**.

## 5.4 ANALYSIS THROUGH CAUSAL INTERVENTION

Because there are two inputs of policy network $\pi_\theta$, we utilize the *intervention* technique in causal inference literature (Rubin, 1974; 1978; Pearl, 1995) to study the causal effect of the coarse action and observation history, which is our core argument.

**Intervention on the coarse action** $\zeta_t$**.** To study the causal effect of the coarse action in our model, we conduct causal intervention experiments on $\zeta_t$ by setting its value manually and keeping all other

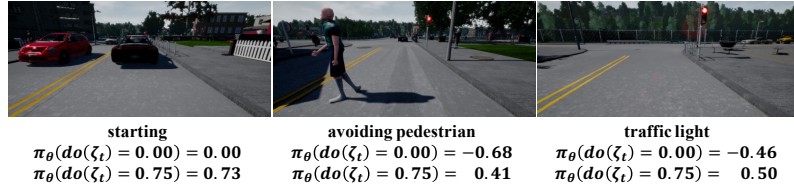

| starting | avoiding pedestrian | traffic light |
|---|---|---|
| $\pi_\theta(do(\zeta_t) = 0.00) = 0.00$ | $\pi_\theta(do(\zeta_t) = 0.00) = -0.68$ | $\pi_\theta(do(\zeta_t) = 0.00) = -0.46$ |
| $\pi_\theta(do(\zeta_t) = 0.75) = 0.73$ | $\pi_\theta(do(\zeta_t) = 0.75) = 0.41$ | $\pi_\theta(do(\zeta_t) = 0.75) = 0.50$ |

Figure 5: Examples of scenarios with high causal effects. The $\pi_\theta(do(\zeta_t) = 0.00/0.75)$ means that we manually set the value of $\zeta_t$ to 0.00 and 0.75 respectively, fix the $\tilde{o}_t$ and then get the output of the $\pi_\theta$. The larger the difference between $\pi_\theta(do(\zeta_t) = 0.00)$ and $\pi_\theta(do(\zeta_t) = 0.75)$, the stronger the causal effect of $\zeta_t$.

factors the same. For easier interpretability, we only intervene on the acceleration dimension, i.e. setting throttle to 0 and 0.75 (the highest value in expert demonstrations), which can be denoted by do-calculus $do(\zeta_t) = 0/0.75$. The causal effect is defined by $CE = |\pi_\theta(do(\zeta_t) = 0) - \pi_\theta(do(\zeta_t) = 0.75)|$, where we omit another input variable $\tilde{o}_t$ for simplicity. Through studying the causal effect of $\zeta_t$ along the trajectories, we find that $\zeta_t$ tends to have a very high causal effect at some critical moments such as the examples in Figure 5, which verifies **H3 (causality correctness)**. Especially, the causal effect of $\zeta_t$ is high when the car is starting, but it decreases after the car is started because at this time it is necessary to refine its actions according to observation history, which is what we expected.

**Intervention on the observation history** $\tilde{o}_t$**.** Similarly, to study the causal effect of the observation history $\tilde{o}_t$, we intervene it by repeating the current frame $H$ times, i.e. $do(\tilde{o}_t) = [o_t, o_t, \cdots]$ which creates counterfactual stationary cases, i.e. the previous action is 0 and we denote it as $\pi(do(\tilde{o})) = 0$. Recall the example in Section 3.2, to investigate what factors the agents use to determine whether to move forward or stop, we count the percentage of model outputs that change from accelerate to stop after we intervene on the input sequence, i.e.

$$\frac{N(\text{speed} > 0, \pi(\tilde{o}_t) > 0, \pi(do(\tilde{o})) = 0)}{N(\text{speed} > 0, \pi(\tilde{o}_t) > 0)}$$

We count this metric for BC-OH and our model on the same dataset. There are 66.43% samples changing from accelerate to stop in BC-OH such as the example in Figure 6, even though there is no signal to stop in the scene, e.g. vehicles, pedestrians, red lights or other obstacles. This illustrates that surprisingly in more than half of the cases, BC-OH is making decisions only according to the previous action and ignores the current scene, which is causally incorrect. In the meanwhile, there is only 27.89% in our model, indicating that our model learns correct causal relation and significantly alleviates the copycat issue, which also verifies **H3 (causality correctness)**.

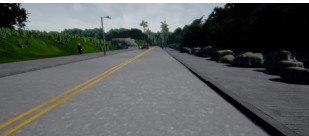

| BCOH: $\tilde{o}_t$: 0.75 | $do(\tilde{o}_t)$: 0.05 |
|---|---|
| Ours: $\tilde{o}_t$: 0.7 | $do(\tilde{o}_t)$: 0.75 |

Figure 6: An example of intervention on $\tilde{o}_t$.

## 6 CONCLUSION

In this paper, we propose a simple but effective model combination approach to resolve the copycat problem. Our method takes the best of BC-SO and BC-OH to predict the precise actions according to observation history while avoiding the copycat problem. We verify our method on autonomous driving and robotics control environments. Extensive carefully designed ablation and analysis experiments verify that coarse-to-fine imitation works by providing an alternate more desirable shortcut to the imitator during training, which supplants the problematic copycat shortcut. Our method outperforms the existing methods and significantly alleviates the copycat problem.

## 7 REPRODUCIBILITY STATEMENT

All our experiments can be easily reproduced according to the implementation and training details discussed in Section 5.1 and Section 8.2. The data collection process of CARLA is described in Section 8.2 and Mujoco's is in Section 8.3. Especially, the CARLA100 is a public dataset released by Codevilla et al. (2019). We will release our code and dataset if our paper is accepted.

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

## 8 APPENDIX

### 8.1 BASELINES

**BC-SO&BC-OH.** As introduced in Section 3.2, BC-SO and BC-OH are naive behavioral cloning methods from single observation and observation history respectively.

**Fighting-Copycat-Agents (FCA).** Wen et al. (2020) proposed to remove the unique information about the previous actions $a_{t-1:t-H}$ from the feature extracted from the observation history to prevent the agent to copy the $a_{t-1:t-H}$, based on adversarial learning.

**KeyFrame.** Wen et al. (2021) analyzed the copycat problem in terms of the imbalance data distribution and proposed a re-weighting method to up-weight the demonstration keyframes corresponding to expert action changepoints.

**History-Dropout.** To address the copycat issue, Bansal et al. (2019) introduced a dropout on the observation history to randomly erase the channels of historical frames. We implement it by applying a Dropout layer (Srivastava et al., 2014) on the historical observations.

**Average-Ensemble.** Average-Ensemble is a commonly used model combination approach. We implement it by averaging the outputs of BC-SO and BC-OH at test time.

**DAGGER.** DAGGER (Ross et al., 2011) is a widely used technique to address the distributional shift issue in behavioral cloning and is thought of as the **oracle** of imitation learning through online query.

### 8.2 ADDITIONAL DETAILS ON CARLA EXPERIMENTS

In Section 5, we briefly introduce the experiment setup of CARLA. More details are introduced below.

**Data Collection.** The CARLA100 dataset (Codevilla et al., 2019) is collected by a PID controller. During collecting, 10% expert actions are perturbed by noise (Laskey et al., 2017a). We use three cameras: a forward-facing one and two lateral cameras facing 30 degrees away towards left or right (Bojarski et al., 2016). Both noise injection and multiple cameras are common data augmentation techniques to alleviate distributional shift in autonomous driving.

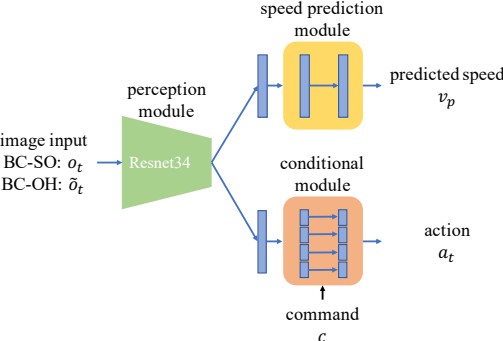

Figure 7: The conditional imitation learning architecture we used as our backbone. The input of BC-SO is the current observation $o_t$ and the input of BC-OH is the observation history $\tilde{o}_t$.

**Architectures.** We use the backbone in conditional imitation learning framework CILRS (Codevilla et al., 2019) as our backbone. The only difference is that our model does not have the input speed (to create a pure POMDP (Wen et al., 2021)). As shown in Figure 7, BC-SO and BC-OH use the same architecture with different inputs, $o_t$ and $\tilde{o}_t$. Illustrated in Figure 8, our method integrates them together by concatenating the output of coarse model (BC-SO) with the features of the penultimate FC layer of policy model (BC-OH). Moreover, the architectures of early fusion model and middle fusion model mentioned in Section5.3 are shown in Figure 9. In particular, early fusion means we concatenate the coarse action $\zeta_t$ with the input images, and middle fusion means that we concatenate it with the output feature of Resnet.

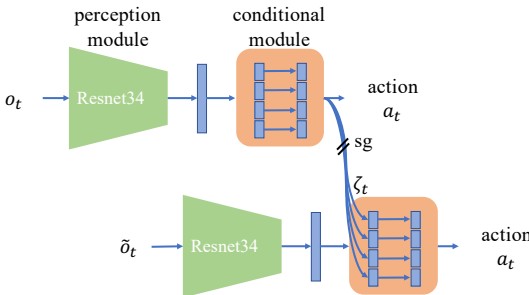

Figure 8: The architecture of our method. For simplicity, we omit the speed prediction module and the input command when drawing the figures, and sg means stop-gradient.

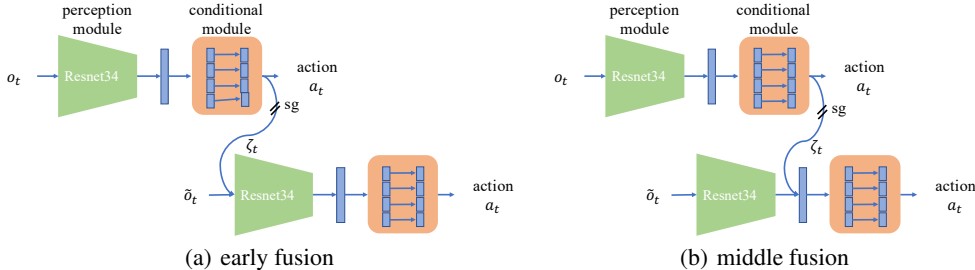

Figure 9: The architectures of the early fusion model and middle fusion we study in Section 5.3.

**Training Details.** We use the $L_1$ loss to train all the models. We use Adam optimizer, set the initial learning rate to $2 \times 10^{-4}$ and decay the learning rate by 0.1 whenever the loss value no longer decreases for 5000 gradient steps. We set the minibatch size to 160 and train all the models until convergence (the learning rate equal to $1 \times 10^{-7}$). Furthermore, we apply several commonly used techniques to our training process. We utilize the noise injection (Laskey et al., 2017a) and multi-camera data augmentation (Bojarski et al., 2016; Giusti et al., 2015) to alleviate the distribution shift in offline imitation learning. All the models use the speed regularization (Codevilla et al., 2019) to address the copycat problem (also called inertia problem in their paper) to some extent. And we use ImageNet pretrained ResNet34 (Deng et al., 2009; He et al., 2016) as the perception module to get a better initialization (Codevilla et al., 2019) and the weighted control loss to balance the models' attention to each action dimension. Furthermore, different from the previous works (Codevilla et al., 2018; 2019; Wen et al., 2021), we use two-dimensional action space $a \in [-1, 1]^2$ for steering and acceleration, where the positive acceleration value means applying throttle and the negative one means braking. We empirically find that using acceleration as output modality is better than predicting throttle and brake separately among all baselines (except FCA) and our method.

## 8.3 Additional Details on Mujoco Experiments

**Data Collection.** We first train a RL expert with PPO (Schulman et al., 2017) and use it to generate expert demonstration by rolling out in the environment. Specifically, we collect 10k samples for Ant and Walker2D, and 20k for Hopper based on imitation difficulty.

**Architectures.** We follow a simple design for network architectures, shown in Figure 10. For both the coarse action and policy model, we use a three-layer MLP network. We concatenate the output action from the coarse action model to the output of policy model, and input that to another fully-connected layer for the final output action.

**Training Details.** We use MSE loss and Adam optimizer to train all models. We use a learning rate of 1e-4 for both HalfCheetah and Ant, and 1e-5 for Hopper and use linear learning rate decay. For each environment, we train it for 1000 epochs until convergence. We set the minibatch size to 64. We

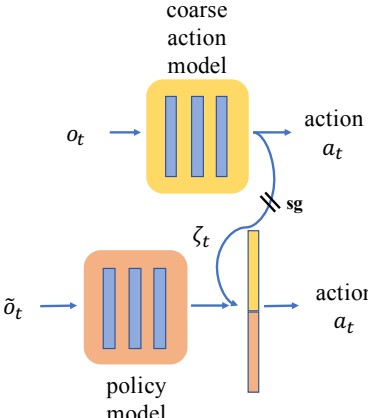

Figure 10: The architecture of our method for MuJoCo experiments, where sg means stop-gradient.

train each method for three times, and report the mean and standard deviation of evaluation rewards for the last three evaluation steps.

