# OpenReview forum: "Fight fire with fire: countering bad shortcuts in imitation learning with good shortcuts"
_ICLR.cc/2022/Conference — ICLR 2022 Submitted_

### Official Review · Reviewer_Lt5X · 2021-11-02

**Correctness:** 3
**Technical Novelty And Significance:** 2
**Empirical Novelty And Significance:** 3
**Recommendation:** 3
**Confidence:** 4

**Main Review:**

[Strength:]
Overall, this paper starts with the "copycat" problem in imitation learning when using historical observations and proposed an ensemble approach to balance between a single observation-based predictor and a history of observations based predictor. As per my knowledge, this combination is novel. Experimental results also demonstrate its effectiveness.

[Weakness:]
The authors did an analysis empirically on why the proposed approach can address the "copycat" problem, however, I feel it still lacks thorough studies regarding what exactly is the "copycat" problem either in a distributional perspective or a decision making perspective. As a result, I think it is insufficient to recognize the authors' novel contributions.


**Summary Of The Paper:**

This paper presents a combined approach that combines two predictors with one based on a single observation (BC-SO) and the other based on a history of observations (BC-OH). The intuition behind this approach is that BC-OH has the "copycat" problem which basically means the predictor is trying to simply extrapolate the previous action instead of using data to do some decision makings, while BC-SO does not have such a problem but suffers from the partial observability issue. The authors also evaluate their method based on the CARLA and MuJoCo continuous control tasks. Results clearly show the effectiveness of this combined approach.

**Summary Of The Review:**

The authors proposed a combined approach to address the "copycat" problem in imitation learning when learning with a history of observations. This approach is new, the problem is interesting but the paper lacks satisfactory investigation of the problem itself and how the proposed method can solve the problem.

---

> ### Author Response · Authors · 2021-11-18
> **Thanks for the feedback and we will improve the writing.**
>
> Thanks for your suggestion.
>
> Our intention is to explore the copycat problem from the shortcut learning perspective (Geirhos et al. 2020) and propose a simple but effective solution by utilizing the correct shortcut. Because there are few accurate and detailed theoretical foundations for shortcut learning in neural networks, we choose to analyze it empirically and design a series of experiments to verify the analysis. Thank you for the feedback, and we will improve the paper to discuss the copycat problem and its connection to shortcut learning more clearly.

---

### Official Review · Reviewer_ESVr · 2021-11-03

**Correctness:** 3
**Technical Novelty And Significance:** 4
**Empirical Novelty And Significance:** 2
**Recommendation:** 5
**Confidence:** 4

**Main Review:**

#### **Strengths**
* Clear to read and their method is succinctly illustrated (Figure 4)

* Motivations from human-decision making and various works in cognitive psychology

* Improvements over previous methods

* Ablation on design choices in their method

#### **Weaknesses**
* I would like to see more insight on why their method increases performance by such a drastic margin. Section 4.3 and Table 3 provide some hints, but I am still not completely sure where the improvements came from. If the copycat problem stems from optimization difficulties and networks converging to the simple solution, why wouldn't the refinement model simply copy the coarse action?

* In the ablations, using BC-OH $\pi_\phi$ (coarse) and BC-SO as $\pi_\theta$ (refinement) leads to a significant drop in performance, performing even worse than the BC-OH baseline in Table 1. I fail to see why this would happen since $\pi_\theta$ could simply copy over the coarse action and would like to see a bit more elaboration from the authors.

#### **Suggestions**

* Elaborating on section 5.4, to see if the refinement model actually refines the prediction, an interesting experiment would be to actually quantify the difference between the coarse action and the final action in each model in the ablation.

* The main results and ablations are performed in CARLA so MuJoCo results can moved to supplementary

* I am familiar with the CARLA NoCrash benchmark and know the dense setting in particularly is prone to a lot of noise in evaluation (traffic jams which cause early termination or timeout). I would like to see the community migrate towards more robust experiment testbeds in CARLA (the more recent online leaderboard and provided routes).


**Summary Of The Paper:**

This work proposes a technique for training imitation learning policies that use temporal information. They show policies trained with their method are less "causally confused" (copycat problem) compared to other behavior cloning methods. In their approach, they decompose a policy into two - the first policy only takes in the current time step's observation (reactive) and outputs an action, along with a intermediate feature (their best performing model just uses the action as the feature). The second policy (refinement stage) takes in the feature, along with the history of observations, and outputs the final action. They evaluate their method in an autonomous driving simulator (CARLA NoCrash benchmark) and MuJoCo environments.

**Summary Of The Review:**

Overall this work addresses a well known problem (copycat problem) and builds towards training better imitation learning policies with temporal information. I enjoyed that the method is simple and effective on the NoCrash benchmark, but I find the justification and analysis a bit incomplete in its current state to be recommended for acceptance.

---

> ### Author Response · Authors · 2021-11-18
> **More explanation about our method and the ablation study results**
>
> Thank you very much for your time and valuable suggestions.
>
> 1. “Why wouldn't the refinement model simply copy the coarse action?”
>
> The refinement model indeed does not simply copy the coarse action predicted from the current observation alone. This is because BC-SO-style coarse action prediction does not yield the lowest imitation error on the training data. As Section 3.2 shows, and prior works addressing this problem have also reported, BC-OH yields substantially lower training errors than BC-SO. In other words, copying the coarse action and ignoring the observation history would lead to higher training errors for the refinement model, compared to utilizing some historical observations to inform and improve its prediction.
>
> 2. “In the ablations, using BC-OH $\pi_{\phi}$ (coarse) and BC-SO as $\pi_{\theta}$ (refinement) leads to a significant drop in performance”
>
> When using BC-OH as $\pi_{\phi}$ and BC-SO as $\pi_{\theta}$, the resulting success rate (31.7%) is only slightly lower than BC-OH (34.1%), which is within the standard deviation. We believe that this difference may be caused by the slightly different architectures. Essentially, both of these two models suffer from the severe copycat problem. Note that these tables report driving performance, and not the imitation error on training data. The training loss of BC-OH is 0.03658 and that of this ablation model is 0.04011, which are comparable.
>
> 3. “I am familiar with the CARLA NoCrash benchmark and know the dense setting in particular is prone to a lot of noise in evaluation”
>
> For the testbeds in CARLA, the evaluation of NocrashDense is indeed noisy. After we submitted the paper, we also conducted experiments in NocrashEmpty and NocrashRegular and we found our method still outperformed all other methods in these benchmarks (results below). We will include these results in the next version.
>
> |                 | NocrashEmpty     | NocrashRegular   |
> | :-------------: | ---------------- | ---------------- |
> |      BC-SO      | 44.9 $\pm$ 6.7     | 37.6 $\pm$ 6.1     |
> |      BC-OH      | 78.4 $\pm$ 11.6    | 67.1 $\pm$ 10.8    |
> |      Ours       | **89.9 $\pm$ 1.4** | **81.1 $\pm$ 4.0** |
> |    Keyframes    | **90.1 $\pm$ 5.7** | 74.4 $\pm$ 7.3     |
> | History Dropout | 85.1 $\pm$ 3.7     | 70.1 $\pm$ 4.0     |
> |       FCA       | 70.4 $\pm$ 7.6     | 58.0 $\pm$ 8.0     |
> |   DAGGER120K    | 83.2 $\pm$ 9.4     | 69.7 $\pm$ 8.4     |

---

### Official Review · Reviewer_AJ5m · 2021-11-06

**Correctness:** 2
**Technical Novelty And Significance:** 3
**Empirical Novelty And Significance:** 2
**Recommendation:** 3
**Confidence:** 4

**Main Review:**

The paper describes a simple solution to a common problem (copycat behavior) in BC. Contributions in this area are important, timely, and will be applicable to many areas. The provided evidence is mostly empirical with a few qualitative arguments about why the approach should be superior to state-of-the-art.

The approach is compared to a number of baselines which is commendable. Unfortunately, it is hard to confirm the results since the code has not been shared at the time of submission.


- While the paper states that both models are trained jointly, I would assume that the BC-SO policy could be trained independently. Once training is complete, the BC-OH policy can then be trained with outputs from BC-SO. It might be worthwhile to state this a bit more clearly since training jointly usually implies that one model propagates through the other, which is not the case here due to the gradient stop.

- What is the imitation loss used in the experiments? Does the choice of losses have an influence on the effectiveness of the presented approach?

- I would imagine that the presented technique depends on the available capacity of the networks. I.e. if the capacity of BC-OH is lower, it will rely more quickly on the output of BC-OS. Could you provide an ablation?

- Much of the explanation about why this approach should work better is fairly philosophical, hard to follow, and quite a few statements made are speculation or questionable.

- To me, the most convincing argument is that the optimization surface becomes easier to train. In principle, I could imagine this to be similar to how resnet architectures benefit from a certain structure. The paper would improve by visualizing this benefit. Please take a look at [1] on how this could look like.

- Most experiments are not statistically significant. 3 seeds are not enough. I would also recommend changing the notation from simply stating the standard deviation to showing the 95% confidence interval. Simply state the bounds of the T-test: /mu +- t * std/sqrt(n) For 3 seeds you would have to multiply each standard deviation with 4.303/sqrt(3)=2.48, for 5 seeds with 2.776/sqrt(5) and for 10 seeds with 2.262/sqrt(10). With the current factor of 2.48, many of the confidence intervals would be overlapping, and the statistical significance of the provided results is not ensured. Since the results seem to be fairly high in variance I would recommend running 10 seeds.


[1] Li, H., Xu, Z., Taylor, G., Studer, C., & Goldstein, T. (2017). Visualizing the loss landscape of neural nets. arXiv preprint arXiv:1712.09913.

**Summary Of The Paper:**

The paper proposes to combine behavioral cloning (BC) from single observations with observations from an observation history to address the copycat problem of behavioral cloning on observation histories alone. The approach first learns a coarse policy that predicts actions from a single observation like it is usually done in single observation BC. They then feed the input into another policy that also receives an observation history which then outputs the combined action.
The approach is evaluated on CARLA autonomous driving from images and various MuJoCo continuous control tasks.


**Summary Of The Review:**

I believe that the paper is interesting (although the approach is extremely simple) and could provide insights into how architectures and engineered biases can be beneficial for behavioral cloning. Unfortunately, the presented results are not statistically significant. Additionally, I believe that the paper could do better in substantiating the claims about why this approach should perform better. This could either be done through loss function visualizations, mathematical arguments (bottlenecks and compression might be applicable), or other quantitative reasoning. I am happy to change my view if these improvements are provided.

---

> ### Author Response · Authors · 2021-11-18
> **Some discussion about the architecture and training process, and a new ablation study result**
>
> We really appreciate your interest and kind suggestions.
>
> 1. “the BC-SO policy could be trained independently”
>
> We performed this ablation experiment where we train BC-SO first. Please refer to “w/o end-to-end” in Table3 In our ablation. Training the BC-SO policy first does indeed yield similar performance to our approach, as the reviewer suggests. We opt for the less cumbersome single-phase end-to-end training strategy in the rest of our experiments, and will make sure to highlight the gradient stop more prominently.
>
> 2. “What is the imitation loss used in the experiments? ...“
>
> We use the L1 loss for CARLA and MSE loss for Mujoco, following the default setting commonly used (Codevilla et al. 2018,2019; Wen et al. 2020,2021; de Haan et al. 2019). Our approach involves an architectural modification, and is not tied to any one choice for imitation loss, and we expect it to be compatible with other losses.
>
> 3. Ablation on the network capacity of BC-OH
>
> We reduce the capacity of $\pi_{\theta}$ by changing the Resnet34 to Resnet18 and reducing the number of neurons in each FC layer to half of the original. The online evaluation success rate is 48.9% while our default model is 49.3%, which are very close. Therefore this ablation experiment proves that our approach is robust under different network capacities.
>
> 4. “Why this approach should work better”
>
> We agree that our approach likely helps to avoid optimization difficulties, as we mention in Section4.3. We empirically analyze our method from the shortcut learning perspective in Section4.3, make three hypotheses about why our method works in the first paragraph in Section5, and then we conduct a series of experiments to verify the hypotheses. Thanks for your suggestion and we will improve our visualization.
>
> 5. “I would recommend running 10 seeds”
>
> To reduce the training and evaluation variance, we re-trained each model for 3 times and re-tested them for 3 times (Wen et al. 2021), so the final result is the average of 9 testing scores. The training and online testing of CARLA are too time-consuming (nearly one day to train and 2-3 days to evaluate), so we don’t have time to train each model 10 times. We will do that later and thanks for your suggestion.
>
> 6. We believe that the simplicity of the approach is actually a key strength, and that we believe our method will be easy to reimplement and evaluate in many other settings.

---

### Official Review · Reviewer_Dq7x · 2021-11-08

**Correctness:** 3
**Technical Novelty And Significance:** 2
**Empirical Novelty And Significance:** 2
**Recommendation:** 3
**Confidence:** 2

**Main Review:**

Strengths:
- Good thorough ablation study to test the combination of two BC methods.
- Good range of simulation based continuous tasks.

Weakness:
- The paper is difficult to follow and I encourage the authors to improve the writing of the paper. Right now it is not in a shape to be published at a top-notch conference (ICLR).
- The first two lines of the abstract is not clear to me. It will be great if you could re-write the abstract to have a better understanding of the paper.
- The idea presented in the paper seems incremental and has less novelty.
- There is no algorithmic complexity of the proposed method in the paper. I highly encourage the authors to present their method in algorithmic form and show its complexity after combining.
- There are a lot of redundant information presented in the paper which could be removed (example problem with BC-SO and BC-OH presented in Intro, Related Work and Preliminaries) and make space for important stuff suggested in above suggestion.
- From section 4.2, the authors have switched the language of BC-single observation to instantaneous observation. Is there a particular reason for change in the language? If not, it would be good to be consistent  which helps the readers to follow easily.
- There is a typo in Section 3.2: Copycat Problem, fourth last line, purposes -> proposes

**Summary Of The Paper:**

The paper addresses a behavioral cloning (BC) distributional shift problem in a POMDP setting. The paper proposes to combine strengths/merits of BC with single observation and BC with historical observation. The author performs an ablation study to verify their proposed method. Also, they test the proposed method on four continuous control task ranging from autonomous driving (CARLA simulation) tasks to Mujoco based tasks.

**Summary Of The Review:**

The ideas presented in the paper seems incremental and the paper is not well written to be accepted at ICLR.

---

> ### Author Response · Authors · 2021-11-18
> **We wil improve the writing in this paper.**
>
> Thanks for your kind suggestion.
>
> We recognize that the writing in this paper can be substantially improved, and we will do this for the next iteration. This includes avoiding inconsistent terms like "single’’ vs "instantaneous’’ observation.
>
> 1. "There is no algorithmic complexity of the proposed method in the paper”: Our method does not lead to any substantially higher algorithmic complexity than BC-OH. The number of parameters of BC-OH is  23,489,161 and that of our method is 46,980,370, so the only increasing complexity is the GPU memory. In terms of the amount of memory and #training iterations, our method and the BC-OH are exactly the same throughout our experiments.

---

### Decision · Program_Chairs · 2022-01-20

**Decision:**

Reject

**Comment:**

The reviewers unisono do not accept the paper, because it is (a) not well-written; (b) experimentally not convincing, but addresses a nice problem.  I suggest that the authors address the issues in a subsequent paper, and resubmit to one of the main conferences.